# Can We Mathematically Spot the Possible Manipulation of Results in Research Manuscripts Using Benford's Law?

**Teddy Lazebnik** [1,*]  **and Dan Gorlitsky** [2]

1   Department of Cancer Biology, Cancer Institute, University College London, London WC1E 6BT, UK
2   Department of Economics, Reichman University, Herzliya 4610101, Israel
*   Correspondence: t.lazebnik@ucl.ac.uk

**Abstract:** The reproducibility of academic research has long been a persistent issue, contradicting one of the fundamental principles of science. Recently, there has been an increasing number of false claims found in academic manuscripts, casting doubt on the validity of reported results. In this paper, we utilize an adapted version of Benford's law, a statistical phenomenon that describes the distribution of leading digits in naturally occurring datasets, to identify the potential manipulation of results in research manuscripts, solely using the aggregated data presented in those manuscripts rather than the commonly unavailable raw datasets. Our methodology applies the principles of Benford's law to commonly employed analyses in academic manuscripts, thus reducing the need for the raw data itself. To validate our approach, we employed 100 open-source datasets and successfully predicted 79% of them accurately using our rules. Moreover, we tested the proposed method on known retracted manuscripts, showing that around half (48.6%) can be detected using the proposed method. Additionally, we analyzed 100 manuscripts published in the last two years across ten prominent economic journals, with 10 manuscripts randomly sampled from each journal. Our analysis predicted a 3% occurrence of results manipulation with a 96% confidence level. Our findings show that Benford's law adapted for aggregated data, can be an initial tool for identifying data manipulation; however, it is not a silver bullet, requiring further investigation for each flagged manuscript due to the relatively low prediction accuracy.

**Keywords:** statistical analysis; anomaly detection; first digit law; results reproduction

## 1. Introduction

The scientific community places great emphasis on maintaining the integrity and dependability of published manuscripts [1–3]. The accuracy and validity of research findings are crucial for advancing knowledge and establishing evidence-based policies [4,5]. Unfortunately, the existence of fraudulent or deceptive research across different disciplines presents a substantial obstacle for scientists [6–8].

There are various motivations behind the presentation of misleading results in academic papers [9]. These motivations range from seeking professional recognition by publishing in high-impact journals to securing funding based on impressive previous work and even attempting to salvage a study that did not yield the desired outcomes [10–12]. Furthermore, the traditional peer review process often fails to identify deliberate attempts at results fabrication, particularly when raw data are not provided, although the absence of raw data itself is an undesirable practice [13–16]. Studies even highlight the potential benefits journals gain from publishing manuscripts containing errors, misconduct, or fraud, such as an increase in their scientometric matrices and popularity [7]. This issue is particularly relevant in the field of economics, where data analysis and statistical properties play a crucial role, but restrictions on sharing raw data, driven by privacy concerns and the protection of business secrets, make it difficult to scrutinize the findings [17]. Consequently,

scholars in this field may find it tempting to manipulate results with minimal risk involved, creating an undesirable environment for research integrity [18–20].

Ensuring the integrity and trustworthiness of research studies is essential, and this necessitates the identification and exposure of potential inconsistencies or intentional misrepresentations within research manuscripts [21]. Traditional methods of detecting anomalies or suspicious patterns often involve a manual examination, which is a time-consuming and resource-intensive process [22,23]. Furthermore, this approach demands a high level of expertise in each respective field, thereby limiting the number of individuals capable of performing such tasks [24]. As a result, there is an increasing demand for objective and automated approaches to assist in the identification of possible falsehoods in academic research, particularly when the original data are unavailable for review [25,26]. To address this gap, previous studies have shown show that one can adopt Benford's law [27], which describes the relative frequency distribution for the leading digits of numbers in datasets, to identify manipulation in data [28,29]. This method assumes that in legitimate data collected through experiments, the distributions of the leading digits of numbers in datasets would follow some distribution. This assumption does not necessarily hold, especially considering the wide range of research topics, data types, and novel experiment methods (such as in silico experiments), making this method useful but limited. Nonetheless, Benford's law has been successfully implemented in previous studies. For instance, Horton et al. [30] used Benford's law for known retracted papers and non-retracted papers from the same realm. The authors experimentally showed that their method could obtain reasonable results in detecting "problematic" data. In the context of economic research, ref. [31] showed that a large proportion of recent economic studies, based on the relatively small sample considered by the author, did not follow Benford's law for the first digit but did for the second digit.

However, the main challenge of using this method at scale is the lack of raw data for many manuscripts, either due to reasonable or misleading reasons [32,33]. To this end, this paper presents an innovative method leveraging Benford's law, which focuses on devising rules for examining standard statistical analyses like mean, standard deviation, and linear regression coefficients, following promising previous results in this direction [34,35]. Namely, we suggest computing the distribution of the leading digits for aggregated data, following an aggregating operator $f$. Since many manuscripts use similar statistical analysis operators (mean, standard deviation, linear regression coefficients) and have to report them in the manuscripts, this method can use the available data to test for Benford's law on the aggregated data.

In order to assess the efficacy of the proposed method, a sample of 100 open-access datasets was obtained. For half of these datasets, we computed the actual statistical values, whereas for the remaining half, we intentionally introduced modifications to these values. The findings demonstrated that our proposed approach successfully predicted the outcomes with an accuracy of 79%. Subsequently, we examined 37 retracted manuscripts (due to data or results manipulation), and the proposed method correctly detected 48.6% of these. Moreover, using synthetic data, we explored the limitations of the proposed method in terms of the available data for analysis and the proportion of manipulated data required for detection. In order to evaluate the state of manipulation in research using the proposed method, we collected data from 100 papers published in the top 10 economic journals within the last two years. Disturbingly, our method detected anomalies in 3% of the papers, attaining a confidence level of 96%.

This paper is organized as follows. Section 2 outlines our adoption of Benford's distribution and the construction of the manuscript test. Section 3 describes the methodology employed to collect and preprocess the data used for our experiments, as well as the analysis itself. Section 4 presents the results of our experiments. Section 5 discusses the implementations of our results, followed by an analysis of the applications and limitations of the study, along with possible future work. Figure 1 provides a schematic view of this study.

**Figure 1.** A schematic view of this study. First, we outline the mathematical framework based on Benford's theory. Next, we outline the data acquisition process for the experiments. Finally, we present the experimental setup and results, including a method validation experiment, sensitivity analysis, and an evaluation of recent economics studies, followed by an analysis of the results and a discussion about their implementations.

## 2. Statistical Operators of Benford's Law

Benford's law describes the expected distribution of the leading digits in naturally occurring datasets [27]. It states that in many sets of numerical data, the leading digits are not uniformly distributed. In fact, they follow a logarithmic distribution, as follows:

$$P(d) = log_{10}(d+1) - log_{10}(d) = log_{10}(1 + \frac{1}{d}),$$

where $d \in \{1, 2, \ldots, 9\}$ indicates the leading digit and $P(d) \in [0, 1]$ is the probability that a number would have $d$ as its leading digit. For instance, the probability of the first digit of some number (in a decimal system) being nine would be $log_{10}(9) = 0.046$. To apply Benford's law in practice, one needs to compare the observed distribution of the leading digits in a dataset to those in Equation (2). Deviations from the expected distribution can indicate potential anomalies, irregularities, or manipulation within the dataset.

Let us consider a set of vectors $V := \{v_i\}_{i=1}^k \in \mathbb{R}^{n \times k}$. Formally, an irregularity test based on Benford's law would return $p$, which is the probability value obtained from the Kolmogorov–Smirnov test [36] between the log distribution obtained by fitting $V$ and the distribution in Equation (2). While other metrics are also used, the Kolmogorov–Smirnov test has been shown to be effective with Benford's law to reduce false positives [37]. In order to perform this test on values obtained from $V$ using operator $o$, one needs to first find Benford's distribution associated with such an operator. Hence, let us consider three common statistical operators: mean, standard deviation, and linear regression coefficients. We chose these operators as they have been previously shown to follow Benford's law [34,35,38]. One can numerically obtain these distributions using the convolution operator [39].

Intuitively, one can derive the adapted Benford's law as follows. First, a set of distributions of size $n \in \mathbb{N}$ satisfying Benford's law with some pre-defined $p$-value according to the Kolmogorov–Smirnov test is generated or collected. Next, for each distribution, the $o$ operator's value(s) of the distribution is computed, resulting in a vector of size $n$ of the $o$ operator's value(s). This vector can be treated as a random variable by itself, which defines a distribution numerically, which is denoted by $\zeta$. Thus, given a new set of values divided into multiple distributions, such as several groups in an experiment, one can assess whether the $o$ operator's value(s) distribution of these data is statistically different from the one in $\zeta$. Figure 2 presents a visualization of this process, where the $o$ operator is set as the mean operator.

Formally, we define an anomaly test to be $T_o(D)$, where $T_o : \mathbb{R}^n \to [0, 1]$ is a function that accepts a vector $D \in \mathbb{R}^n$ and an operator $o$ and returns a score of the probability that $D$ is an anomaly with respect to the operator $o$. Formally, for our case, we associate each operator $o$ with its Benford's distribution, and $T_o(D)$ is implemented to return $1 - p$, where $p$ is the probability value obtained using the Kolmogorov–Smirnov test [36] between the distribution associated with the operator $o$ and the same one after fitting to $V$.

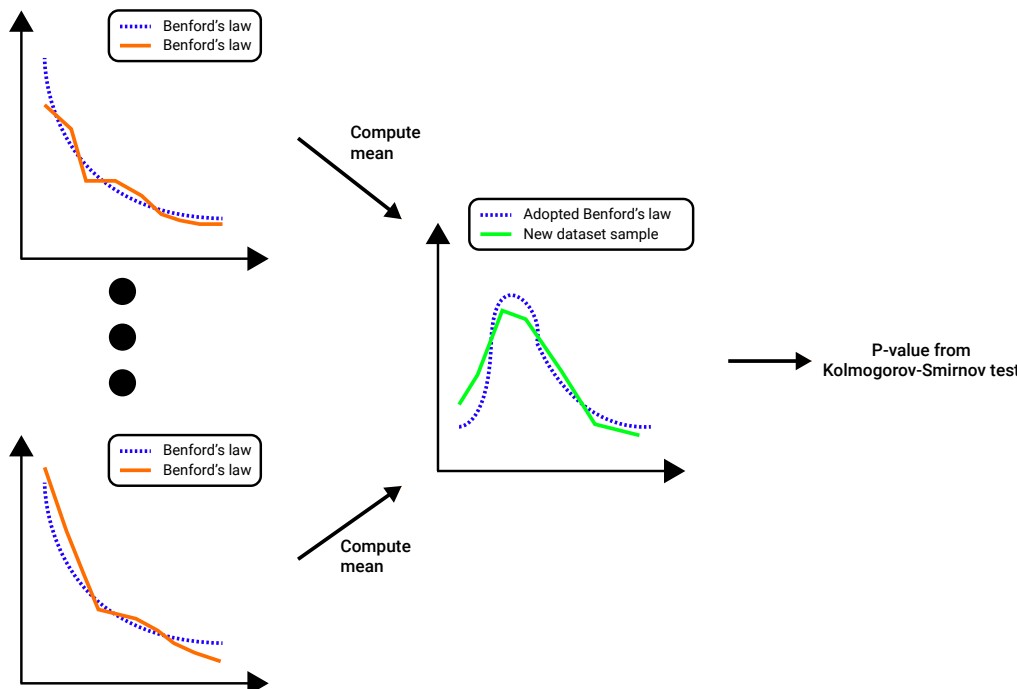

**Figure 2.** An example of the mean operator of the adapted Benford's law obtained numerically.

To practically compute the operators' distributions, for each operator, we numerically generated 1000 random samples and calculated the results for each of them. For this process, we set $p = 0.05$ for the Kolmogorov–Smirnov test's $p$-value. We denoted the worst result obtained as $a \in [0, 1]$. In order to ensure that the proposed test could numerically produce results in the range of $[0, 1]$, for each outcome, $x$, we computed and reported $(x - a)/(1 - a)$. Importantly, for each operator, the obtained Benford's law requires the manual definition of a threshold that determines whether a distribution is similar enough to the expected Benford's distribution or not. To this end, we obtained the threshold for each operator manually using a grid-search approach [40] and then choosing the threshold that resulted in the highest accuracy.

## 3. Experimental Setup

In this section, we outline the experiments conducted in this study. The first and second experiments were designed to numerically validate the performance of the proposed method. The first experiment used legitimate manuscripts that were manipulated by us, whereas the latter focused on manuscripts that were retracted due to data or results manipulation. Next, we explored the sensitivity of the proposed model in terms of the data size, as well as the proportion of observations manipulated. Finally, we evaluated the number of irregularities in recent academic economic studies to determine possible manipulations. We implemented the experiments using the Python programming language [41] (Version 3.7.5). We set $p \leq 0.05$ to be statistically significant.

### 3.1. Performance Evaluation

For the method's performance validation, we manually collected 100 numerical datasets from Data World (https://data.world/datasets/economics; accessed on 24 October 2023) and Kaggle (https://www.kaggle.com; accessed on 24 October 2023), following [42]. The datasets were randomly chosen from a broad range of fields and represented a wide range of computational tasks. Importantly, we ensured that the datasets were not constrained (e.g., a survey with possible answers ranging from 1 to 5) as these are known to be inappropriate for analysis using Benford's law-based methods [30]. Each dataset was represented by a matrix $D$. We defined a feature $f_j$ of dataset $D$ as follows: $f_j := \forall i \in [1, \ldots, n] : d_{i,j}$. A feature was used to calculate the unitary statistical properties.



Based on these data, for each dataset ($D$) and statistical operator ($o$), we computed $T_o(D)$, obtaining a vector of results denoted by $u$. The overall anomaly probability prediction was defined as $\frac{1}{|u|} \sum_{i=1}^{|u|} u_i$. For half of the datasets, we introduced uniformly distributed noise, which was between 1 and 10 percent of the mean value in a uniform manner. As such, these datasets were not expected to agree with Benford's law; therefore, if the proposed method predicts that they do, it indicates an error. As such, we utilized 50 positive and 50 negative examples. All datasets, both before and after manipulation, are available as supplementary materials.

In addition, inspired by the work of [30], we also explored the retracted papers of Professor James Hunton. Namely, we reviewed and manually extracted aggregated data from $n = 37$ manuscripts authored by Professor James Hunton that had been retracted because there were grave concerns that they contained misstated or manipulated datasets. Since the manuscripts were defined as having been manipulated, it meant that at least one result should indicate manipulation, not necessarily all reported results in all manuscripts. Therefore, for each manuscript, if the model predicted manipulation in at least one of the presented aggregated results (in the form of mean, standard deviation, or linear regression coefficients), the entire manuscript was regarded as manipulated.

### 3.2. Sensitivity Analysis

In order to explore the sensitivity of the proposed method, we estimated two parameters directly influencing the performance of the proposed model: the number of observations and the proportion of the observations manipulated. For both experiments, we initially generated $x \in \mathbb{N}$ sets that comply with Benford's law according to a Kolmogorov–Smirnov test with $p < 0.05$. The numbers ranged from $a \in \mathbb{R}^+$ to $b \in \mathbb{R}^+$, with $a < b$, and were uniformly sampled between -100 and 100 in order to ensure a diverse range of possible values. Afterward, a portion $y$ of these observations, uniformly distributed across the $x$ different sets, were randomly changed by introducing a Gaussian noise with a mean value of 10% of the data's mean value and a standard deviation of 5% of the data's standard deviation. From these data, we computed the $o$ operator values for each one of the three operators and evaluated, using the proposed method, whether the data had been manipulated, reporting the mean accuracy for $n = 1000$ attempts.

### 3.3. Economic Manuscripts Use Case

For the economic manuscript evaluation, we collected a sample of 100 papers published in 10 leading economic journals over the past two years. These papers served as the test subjects for applying our proposed method to detect anomalies or irregularities. We chose these amounts and distribution to balance the time and resource burden and the statistical power of the sample. In order to use the leading journals in the economics field, we used the Scimago Journal and Country rank website (https://www.scimagojr.com/; accessed on 24 October 2023), using the search terms "economics and econometrics". We ultimately chose the following top 10 journals [43–45]: Quarterly Journal of Economics, American Economic Review, Journal of Political Economy, Journal of Finance, Review of Economic Studies, Econometrica, Journal of Economic Literature, Review of Financial Studies, Journal of Marketing, and Journal of Financial Economics. For each journal, we mainly counted how many manuscripts the journal had published in the last two years and then asked the computer to randomly pick 10 indices. Once the indices were obtained, we downloaded these manuscripts from the journals' websites. Next, we manually extracted the results from the manuscripts presented in either tables or figures. For each table or figure, where applicable, we applied our adapted Benford's law. A manuscript was considered manipulated if at least a single table or figure was predicted as having been manipulated.

## 4. Results

In this section, we present the results of the three types of experiments conducted using the proposed method. First, we outline the method's performance evaluation for both the synthetically manipulated and retracted cases. Second, we present the sensitivity analysis of the number of observations, as well as the proportion of observations manipulated, based on synthetic data. Lastly, we present the results of exploring recent economic publications using the proposed method. After obtaining the distributions for all three operators, we found that the optimal threshold values for the Kolmogorov–Smirnov test of the mean, standard deviation, and linear regression coefficients were 0.08, 0.06, and 0.10, respectively. These values were obtained through a grid search ranging from 0.01 to 0.20 with a step size of 0.01.

### 4.1. Performance Evaluation

For the first experiment, we used 50 datasets without manipulation, resulting in a 100% alignment with the standard Benford's law. Additionally, we used 50 datasets with manipulation, which caused them not to align with Benford's law within a 5% confidence interval. Overall, three operators were examined: the mean, standard deviation, and linear regression coefficients. For the first group, the analysis was conducted on $17.3 \pm 5.8$, $13.1 \pm 3.8$, and $21.9 \pm 8.6$ data points, respectively. Similarly, for the latter group, the analysis was conducted on $18.1 \pm 6.3$, $12.4 \pm 3.7$, and $19.5 \pm 8.1$ data points, respectively.

To assess the performance of the proposed method, we evaluated the confusion matrix for the synthetically manipulated dataset, as presented in Table 1. The obtained results indicate an accuracy of 0.79 and an $F_1$ score of 0.77. Notably, the model exhibited a tendency to incorrectly predict manipulation-free manuscripts, identifying seven manipulation-free manuscripts as containing manipulations. Conversely, it also misclassified 14 manuscripts with manipulations as manipulation-free. However, from the perspective of the journal, it is preferable for the model to err on the side of caution by falsely predicting manuscripts as manipulation-free, as falsely accusing innocent authors of results manipulation is deemed more undesirable than missing manuscripts with actual manipulations.

**Table 1.** A confusion matrix of the proposed method on the open-source datasets with synthetic data manipulation.

|  | Positive | Negative | Total |
|---|---|---|---|
| **Positive** | 43 | 7 | 50 |
| **Negative** | 14 | 36 | 50 |
| **Total** | 57 | 43 | 100 |

Similarly, in order to measure the proposed method's performance, we computed the confusion matrix for the 37 retracted manuscripts, as presented in Table 2. The obtained results indicate an accuracy of 0.48.

**Table 2.** A confusion matrix of the proposed method on the open-source datasets.

|  | Positive | Negative | Total |
|---|---|---|---|
| **Positive** | 18 | 19 | 37 |
| **Negative** | 0 | 0 | 0 |
| **Total** | 18 | 19 | 37 |

### 4.2. Sensitivity Analysis

Figure 3 presents a sensitivity analysis for the proposed model in terms of the model's prediction accuracy with respect to the number of observations, as indicated on the *x*-axis, and the proportion of manipulated observations, as indicated on the *y*-axis. The results are presented as the mean value of $n = 1000$ cases and are divided into the mean, standard

deviation, and linear regression coefficient operators. One can observe that all three operators follow similar patterns overall: as the number of observations increases, as well as the proportion of manipulated observations, the model's prediction accuracy also increases. In fact, using the SciMED symbolic regression model, we show that for the mean value, a linear regression best describes the numerical data with a coefficient of determination of $R^2 = 0.71$, whereas the standard deviation and linear regression coefficients are better fitted by second-order polynomials with $R^2 = 0.79$ and $R^2 = 0.83$, respectively. Specifically, the obtained symbolic equations are $acc = 0.046 + 0.032x + 0.051y$, $acc = 0.032 + 0.022x + 0.039y + 0.0039x^2 + 0.0021y^2$, and $acc = 0.019 + 0.009x + 0.027y + 0.004x^2 + 0.007y^2$, respectively, where $acc$ is the model's prediction accuracy, $x$ is the number of observations, and $y$ is the proportion of manipulated observations.

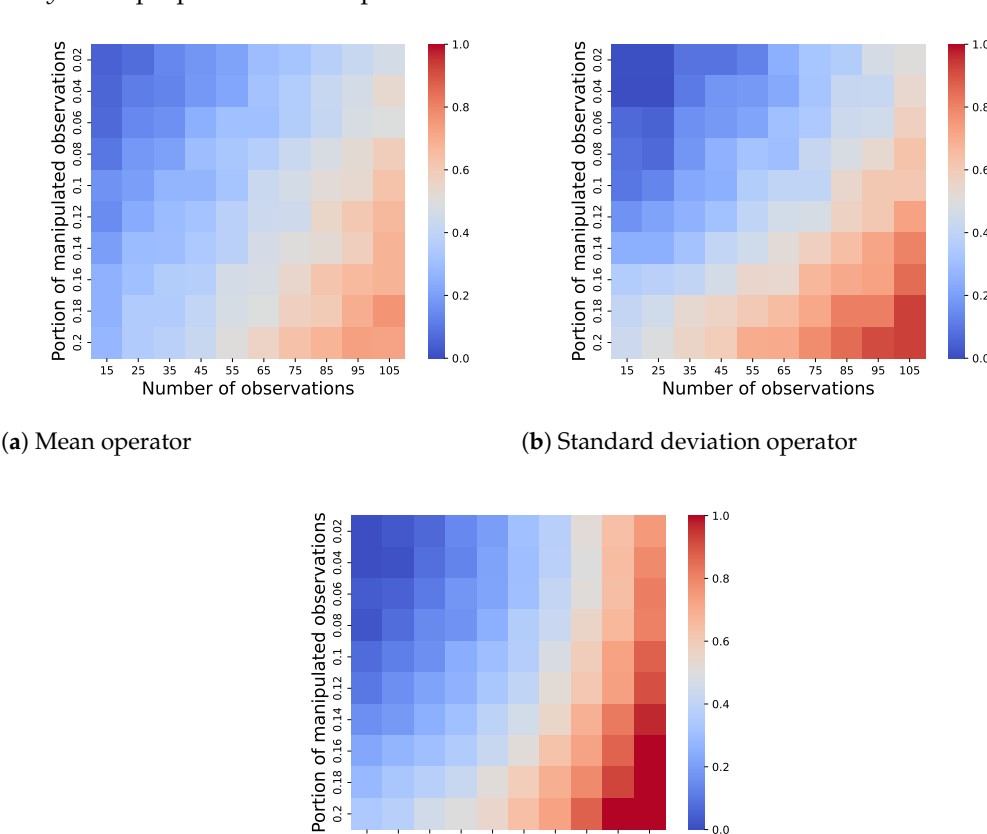

(**a**) Mean operator

(**b**) Standard deviation operator

(**c**) Linear regression coefficient operator

**Figure 3.** Sensitivity analysis of the proposed model in terms of the model's prediction accuracy with respect to the number of observations (*x*-axis) and the portion of manipulated observations (*y*-axis). The results are presented as the mean value of $n = 1000$ cases.

*4.3. Economic Manuscripts Use Case*

Table 3 provides an overview of the predicted number of economic manuscripts flagged as containing results manipulations based on varying confidence levels. It is evident that as the confidence level increases, the number of flagged manuscripts decreases. This observation aligns with expectations since the null hypothesis assumes that the manuscripts are manipulation-free. Hence, higher confidence levels necessitate stronger statistical evidence of manipulation for a manuscript to be flagged.

**Table 3.** The number of manuscripts flagged as containing irregularities according to our method as a function of the required confidence level.

| Confidence level | 90% | 92% | 94% | 96% | 98% |
|---|---|---|---|---|---|
| Flagged manuscripts | 12 | 8 | 6 | 3 | 2 |

## 5. Discussion and Conclusions

In this study, we introduced an innovative approach to identifying potential falsehoods in research manuscripts by applying an adapted version of Benford's law to statistically aggregated data including mean, standard deviation, and linear regression coefficients reported in manuscripts, reducing the need to obtain the raw data used in academic studies. This study continues an extensive study of the adoption of Benford's law for manipulated data detection in academic studies [34,46]. This study's novelty lies in the design of a numerical method to use Benford's law for statistical results reported in manuscripts rather than on raw data, which are often not available.

In order to evaluate the efficacy of our method, we conducted two experiments, which showed that the proposed method could detect data manipulations to some extent. To be exact, Table 1 shows that the proposed method achieved an accuracy of 0.79 and an $F_1$ score of 0.77, indicating its ability to identify potential anomalies, albeit with some limitations. In particular, the method identified manipulation-free manuscripts as manipulated 14% of the time, which may result in innocent authors being falsely blamed, causing far-reaching challenges for journals, academic institutes, and scholars. This outcome can be partially explained in cases where the sample size was small, which is more likely to "confuse" the Kolmogorov–Smirnov test, causing it to predict that data that actually follow Benford's law do not do so and resulting in false positive predictions. This outcome was further supported by the fact the proposed method correctly detected 48.6% (18/37) of the manuscripts predefined as manipulated, with a 95% confidence level, as presented in Figure 2. This indicates the proposed model was able to detect manipulation, as previously reported. However, this is far from optimal and cannot be used as a final decision-making mechanism, but rather as an alerting system. Furthermore, a more sophisticated approach is required [46,47]. Interestingly, previous attempts on the same restricted manuscript datasets, achieved over 70% accuracy [30]. However, the authors used the raw data and as many as three digits after the decimal point. The fact that they used a much richer and extended representation of these manuscripts explains the difference in the methods' performances. That being said, the latter method is much more limited when the raw data are unavailable or only a small number of digits after the decimal point are present [48]. In a more generic sense, compared to Benford's law-based methods, these outcomes highlight a trade-off commonly found in statistical tests, where methods that require fewer data usually also provide less accurate results.

In addition, when considering the sensitivity of the model to the number of available observations, an expected pattern emerged in all three tested operators, where having more observations resulted in better prediction accuracy. Similarly, when more observations were manipulated, the easier it was for the model to detect manipulation, even with a small number of observations. In particular, it seems that linear regression was the most sensitive to the number of manipulated observations. For a large enough number of observations (i.e., over 15 for our analysis), it is a promising candidate for analysis. In comparison, the mean operator required around 20% more observations to obtain similar results to the two other operators. This outcome can be attributed to the fact it is much simpler compared to the other two methods, while also reducing "noise" linearly in relation to the number of observations, which can "hide" manipulations in the raw data. Interestingly, although previous research has shown that for classical Benford's law, 80 observations are required to efficiently test whether data follow Benford's law [49], our results indicated that even small sample sizes of around 17, 13, and 28 observations are efficient for the mean, standard deviation, and linear regression operators, aligning with [30], which suggests

that 22 observations are enough to apply Benford's law when multiple digits are present. This indicates that while the simplistic usage of Benford's law requires more data, a more sophisticated usage reduces this requirement while producing comparable results.

Building upon this premise, our experiment on recent economic data, which are commonly known to follow Benford's law, reveals alarming findings, as approximately 3% of the manuscripts exhibited anomalies, inaccuracies, or even explicit manipulations, with a confidence level of 96%. These outcomes unfortunately align with existing trends in academic fraud practices, underscoring the significance of our approach in uncovering inconsistencies and deliberate misrepresentations in academic research. Furthermore, these numbers agree with self-confessed fraud rates among academic economists of 4% [50,51]. Although this is not a strong indication by itself, as these predictions were not validated, it shows that the model predicted a similar order of magnitude to an established one, further supporting the fact that it is useful.

By leveraging Benford's law, our method offers an objective and automated solution to complement traditional manual scrutiny. Furthermore, it holds particular relevance in fields like economics, where researchers heavily rely on data analysis and statistical properties, often lacking access to raw data due to privacy or proprietary constraints. While our results demonstrate the promise of our approach, there are several limitations to consider. First, our method relies on the assumption that the reported aggregated data follow Benford's distribution, which may not always hold [52,53]. Second, our approach requires the development of tests for each statistical operator, making it hard to utilize a wide spectrum of fields and manuscripts that may use and report a large number of different statistical analysis methods. Third, our method does not provide definitive proof of fraud or misconduct but rather serves as a sign of potential irregularities that warrant further investigation, thus only slightly reducing the time and resources required for the task. Finally, the publication of this study reduces its effectiveness, as malicious scholars would be aware of the proposed method and develop counter-strategies to overcome it, which is common in other fields like cybersecurity [54].

**Author Contributions:** T.L.: Conceptualization, Data Curation, Methodology, Software, Formal analysis, Investigation, Resources, Writing—Original Draft, and Writing—Review and Editing. D.G.: Methodology, Validation, Investigation, and Writing—Review and Editing. All authors have read and agreed to the published version of the manuscript.

**Funding:** This research did not receive any specific grant from funding agencies in the public, commercial, or not-for-profit sectors.

**Institutional Review Board Statement:** Not applicable.

**Informed Consent Statement:** Not applicable.

**Data Availability Statement:** The code and data that have been used in this study are available upon request from the authors.

**Conflicts of Interest:** The authors declare no conflict of interest.

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
