# Peer review of "Can We Mathematically Spot the Possible Manipulation of Results in Research Manuscripts Using Benford’s Law?"

_data, 2023_

Round 1

Reviewer 1 Report (Previous Reviewer 1)

Comments and Suggestions for Authors

I have checked the revised version carefully. I am happy with the corrections in the revised manuscript. It was improved. So, the current version of this manuscript is suitable for publication. 

Good Luck.

Author Response

We would like to thank the reviewer for the important comments in the previous round and the kind words in the current round

Reviewer 2 Report (Previous Reviewer 2)

Comments and Suggestions for Authors

The authors have addressed most of the comments from the previous round, and there is a significant improvement. However, there are two comments that are not fully addressed, specifically,

Comment 3. While more experiments are included, the questions are not answered properly, such as why the model tends to predict manipulation-free manuscripts incorrectly, how the model is trained, and what anomaly detection threshold is computed. The analysis and explanation need improvement.

Comment 4 is partially addressed. An experiment is conducted using the retracted papers, but the performance is not that promising. More explanation and discussion could make the proposed method more persuasive.

Comments on the Quality of English Language

The quality of the English language is fine.

Author Response

Comment 1: "Comment 3. While more experiments are included, the questions are not answered properly, such as why the model tends to predict manipulation-free manuscripts incorrectly, how the model is trained, and what anomaly detection threshold is computed. The analysis and explanation need improvement."

Answer 1: Thank you for this comment. Let us address each sub-question separately. First, the model predicts manipulation-free manuscripts incorrectly since it is basically solving binary classification task which has false positives. In more detail, usually, for small sample sizes, the KS test we used can confuse a sample to be different from the adoptive Benford's law which would result in false positives. Regarding the model "training". It is important to note that the term "training" is commonly used for data-driven models, in general, and machine learning-based models, in particular. However, in the proposed case, we are using a more classical statistical test so the term "learning" may be misleading. Now, for the process itself, we would like to cite from the manuscript: "Formally, we define an anomaly test to be \(T_{o}(D)\) where \(T_{o}: \mathbb{R}^n \rightarrow [0, 1]\) is a function that accepts a vector \(D \in \mathbb{R}^n\) and an operator \(o\) and returns a score of the probability \(D\) is anomaly with respect to operator \(o\). Formally, for our case, we associate each operator \(o\) with its Benford's distribution and \(T_{o}(D)\) is implemented to return \(1 - p\) where \(p\) is the probability value obtained from the Kolmogorov-Smirnov test \cite{kol_siv_test} between the distribution associated with the operator \(o\) and the same one after fitting to \(V\). 

To practically compute the operators' distributions, for each operator, we generated numerically 1000 random samples and calculated the results for each one of them. For this process, we set \(p = 0.05\) for the Kolmogorov-Smirnov test's \(p\)-value. We denoted the worst result obtained as \(a \in [0, 1]\). In order to ensure that the proposed test numerically produces results in the range \([0, 1]\), for each outcome, \(x\), we compute and report \((x-a)/(1-a)\). Importantly, for each operator, the obtained Benford's law requires to manual definition of a threshold that determines if a distribution is similar enough to the expected Benford's distribution or not. To this end, we obtain the threshold for each operator manually using a grid-search approach, choosing the threshold that results in the highest accuracy. " Regarding the actual threshold found for each operator, the mean, STD, and linear regression coefficients have threshold values of 0.08, 0.06, and 0.10, respectively. Following this comment, we altered the text in the Results and Discussion sections to better outline these points. 

Comment 2: "Comment 4 is partially addressed. An experiment is conducted using the retracted papers, but the performance is not that promising. More explanation and discussion could make the proposed method more persuasive."

Answer 2: Thank you for this comment. Indeed, as the reviewer stated, we conducted the requested experiment and reported the results. We agree with the reviewer that the accuracy of the model is sub-optimal and we also acknowledge this in the manuscript itself. To address the query in full, compared to previous approaches in this line of work, the obtained results for this experiment highlight a trade-off. Namely, the proposed method does not require the tester to have the raw data (which is not always available) while it provides less accurate results in return. This trade-off is not surprising as basically, all statistical tests follow similar dynamics in some sense.  Nonetheless, following this comment, we make sure this point is clearly explained in the Discussion section.

------------------------

Please find the modifications in the text presented in bold font. 

Round 2

Reviewer 2 Report (Previous Reviewer 2)

Comments and Suggestions for Authors

The authors have addressed my comments from the previous round. I would suggest accepting the paper, if the authors' replies can be included in the final version.

This manuscript is a resubmission of an earlier submission. The following is a list of the peer review reports and author responses from that submission.

Round 1

Reviewer 1 Report

Comments and Suggestions for Authors

Review Report on: “Can We Mathematically Spot Possible Manipulation of Results in Research Manuscripts Using Benford's Law?

This paper utilizes an adaptive version of Benford’s law, a statistical phenomenon that describes the distribution of leading digits in naturally occurring datasets, to identify potential manipulation of results in research manuscripts, solely using the aggregated data presented in those manuscripts. I doubt that this work's novelty is enough for a top journal as "data" journal.  In order to improve the presentation of the manuscript, maybe the authors want to consider the following comments:

1.     The sections of “Abstract and Conclusions” are very poor, the authors should rewrite these sections.

2.     What does this study add to the literature about Benford's Law? I think the results of this study are expected.

3.     Every section of the manuscript must be written scientifically according to the published literature with appropriate references.

4.   The paper has no scientific results, only overviews.

=================================

Reviewer 2 Report

Comments and Suggestions for Authors

Summary

This paper addresses an important research question which is to verify the validity of the datasets used in research papers based on Benford’s law. Overall, the paper is well organized and written with some interesting experimental results. However, there are some issues that should be addressed before being accepted in the current form.

Comments:

1. In the introduction section, the assumption that Benford's law can be used as a reference to detect dataset validity should be justified more clearly. Do the datasets studied share similar patterns, such as the power law, that make the law adaptive? Some intuition would help verify the correctness of the assumption.

2. In the methodology section, there are some unclear technical details, such as what specific unitart features $f$ are used, and what properties do they capture? Also, why do the distributions of these feature values align with the distribution in Equation 1? 

3. As an experimental study, it is recommended to provide more details for the experimental results. There are some questions that remain unanswered, such as why the model tends to predict manipulation-free manuscripts incorrectly, and how is the model trained and what anomaly detection threshold is computed. The current experiment section (section 4) is brief and the analysis is superficial. 

4. To show the impact of the proposed method, the second experiment can be conducted on datasets used in retracted papers. It is hard to tell if the findings of that experiment are valid or not. More details revealed in this case study could make the proposed method more persuasive.

5. There are some minor grammar/expression issues in the paper, such as "the obtained Benford’s law requires to manual definition of a threshold" on page 3. A careful round of proofreading is suggested.

Comments on the Quality of English Language

The English quality is fine, there are some minor grammar/expression issues in the paper.